# HIERARCHICAL BAYES AUTOENCODERS

## ABSTRACT

Autoencoders are powerful generative models for complex data, such as images. However, standard models like the variational autoencoder (VAE) typically have unimodal Gaussian decoders, which cannot effectively represent the possible semantic variations in the space of images. To address this problem, we present a new probabilistic generative model called the *Hierarchical Bayes Autoencoder (HBAE)*. The HBAE contains a multimodal decoder in the form of an energy-based model (EBM), instead of the commonly adopted unimodal Gaussian distribution. The HBAE can be trained using variational inference, similar to a VAE, to recover latent codes conditioned on inputs. For the decoder, we use an adversarial approximation where a conditional generator is trained to match the EBM distribution. During inference time, the HBAE consists of two sampling steps: first a latent code for the input is sampled, and then this code is passed to the conditional generator to output a stochastic reconstruction. The HBAE is also capable of modeling sets, by inferring a latent code for a set of examples, and sampling set members through the multimodal decoder. In both single image and set cases, the decoder generates plausible variations consistent with the input data, and generates realistic unconditional samples. To the best our knowledge, Set-HBAE is the first model that is able to generate complex image sets.

## 1 INTRODUCTION

Autoencoders are a crucial generative modeling architecture for many applications involving complex data, including image (e.g., Razavi et al., 2019) and scene synthesis (Eslami et al., 2018), image compression (Minnen et al., 2018), video prediction (Lee et al., 2018), and model-based reinforcement learning (Ha & Schmidhuber, 2018). The variational approach exemplified by the VAE (Kingma & Welling, 2013) has the benefit of tractable inference and uncertainty estimation while also allowing reasonable control over bottleneck representation and reconstruction quality. While a number of variants have been proposed to incorporate expressive priors for the VAE latent distribution, such as mixtures of Gaussians (Dilokthanakul et al., 2016) and discrete formulations (Rolfe, 2016), the VAE decoder is very limited given that data dimensions are decoded as conditionally independent Gaussian distributions. VAE samples therefore tend to be blurry and exhibit little meaningful variation around the predicted mean.

On the other hand, complex multimodal decoder distributions can be learned by alternative generative models, including GANs (Goodfellow et al., 2014) and autoregressive models (e.g., pixelCNN) (van den Oord et al., 2016). Conditional formulations of these models (Mirza & Osindero, 2014; Van den Oord et al., 2016) allow controlling the sampling behavior to be class or embedding specific. Typically a known class label or metric embedding is given when training these conditional models. Conditioning on a class label or embedding allows, for instance in the case of face images, to generate a range of different images of a given person identity varying in pose, lighting, expression, etc., by conditioning on the latent embedding learned by a face recognition network (e.g., Van den Oord et al., 2016). These models can be difficult to train to ensure that the decoder pays attention to the conditioning variable.

Encoder-decoder architectures such as VAE-GAN (Larsen et al., 2015) and BiGAN (Donahue et al., 2016) improve sample and reconstruction sharpness through their use of adversarial losses. However, none of these models incorporate multimodal decoders into their formulations, and therefore do not lead to diverse sampling behavior.

In this work we introduce the Hierarchical Bayes Autoencoder (HBAE). The model incorporates a multimodal decoder, in the form of an EBM, which we show under mild assumptions can be trained similarly to a conditional GAN, where the conditioning is the latent code of a VAE. We show that the model is capable of learning to generate crisp and varied samples. Empirically, the stochastic reconstructions produced by the decoder retain semantic features of the respective input examples as verified by distance in Inception feature space, while unconditionally generated samples are diverse and cover the data manifold, as verified with FID scores (Heusel et al., 2017).

We also introduce a simple extension called the Set-HBAE where each data point is a collection of examples (e.g., set of images). In this scenario, the traditional VAE is no longer applicable, because of the inherent one to many mapping between the latent code and the set elements. We show that Set-HBAE is able to be successfully applied to such a setting, and learns to reconstruct input sets in a semantically meaningful way, as well as generate new sets unconditionally by sampling from the latent code prior. Furthermore, we show that the Set-HBAE works well as a method for few-shot classification on ShapeNet (Chang et al., 2015) relative to a supervised baseline, especially in the small data regime.

To summarize, the contributions of this work are:

1. A novel probabilistic generative model trained using variational inference, with a flexible learned multimodal decoder in the form of an EBM.

2. Empirical results showing that the multimodal decoder is able to produce stochastic reconstructions varying around the local data manifold of examples, and diverse unconditional samples.

3. An extension of the HBAE formulation to model sets of inputs, allowing a finer degree of semantic control over sampling behavior, and a new method for few-shot classification of sets.

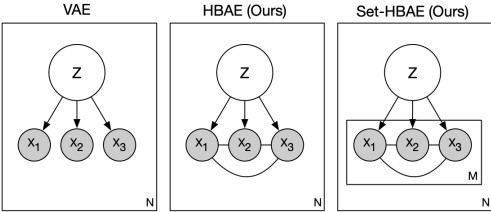

Figure 1: Left: Graphical model for a VAE for sampling $N$ data points, where each sample $\mathbf{x}$ is conditioned on a latent code $Z$ sampled from a prior, and data dimensions $\mathbf{x}_1, \mathbf{x}_2, \mathbf{x}_3$ are conditionally independent given $\mathbf{z}$. Middle: In the HBAE, the decoder distribution $p_\theta(\mathbf{x}|\mathbf{z})$ is replaced by a deep EBM, where data dimensions can interact via undirected connections. Right: In the Set-HBAE, we first sample $N$ set codes $\mathbf{z}$, and we then sample $M$ set elements by repeatedly generating from the EBM conditioned on the set latent code.

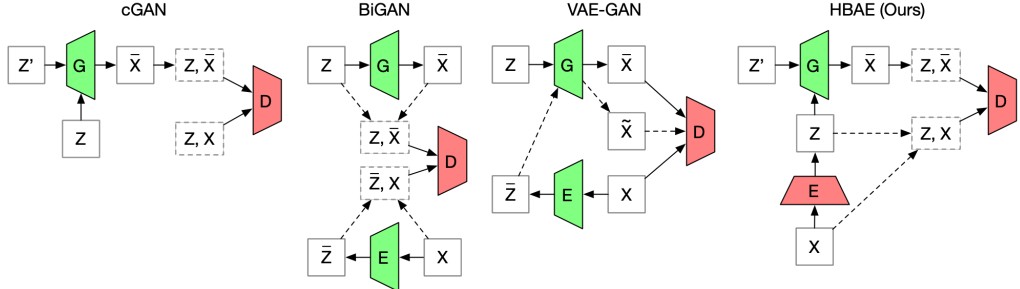

Figure 2: Architectures for different models. For each architecture, we use same colors to group together modules that contribute cooperating terms to the adversarial loss formulation. The HBAE can be interpreted as a cGAN with a VAE encoder where all modules are learned jointly.

## 2 METHODOLOGY

### 2.1 VAEs

VAEs (Kingma & Welling, 2013) represent a family of deep directed graphical models. Given an input $\mathbf{x} \in \mathcal{X}$, a VAE formulates a joint distribution of $\mathbf{x}$ and a latent variable $\mathbf{z} \in R^l$ as

$$p_\theta(\mathbf{x}, \mathbf{z}) = p_\theta(\mathbf{z})p_\theta(\mathbf{x}|\mathbf{z}), \tag{1}$$

where $\theta$ denotes the set of trainable parameters; $p_\theta(\mathbf{z})$ and $p_\theta(\mathbf{x}|\mathbf{z})$ denote the prior and decoder distribution, respectively. As a common design choice, $p_\theta(\mathbf{z})$ is usually set as a simple low dimensional distribution (e.g., an Isotropic Gaussian) which can be easily sampled from; $p_\theta(\mathbf{x}|\mathbf{z})$, referred to as the decoder, is parameterized as a Gaussian distribution with a learned mean $\mu(\mathbf{z}; \theta)$ and an identity covariance matrix. This popular instantiation of VAEs offers a simple objective function which corresponds to a regularized autoencoder where reconstructions are penalized with $\ell_2$ loss.

We argue that such a unimodal decoder is not optimal for at least two scenarios. Firstly, in the case where the dimensionality of $\mathbf{z}$ is much lower than the inherent dimensionality of the input $\mathbf{x}$, a unimodal decoder becomes the bottleneck of learning. In an extreme case, if the latent space is limited to contain only $k$ bits, e.g., a categorical VAE as in Jang et al. (2016) with $2^k$ latent units, with $k$ being a small quantity, the model is forced to assign multiple inputs to any given latent code $\mathbf{z}$. This makes having a multimodal decoder essential to effectively fit the data distribution. Secondly, for settings where the input does not have a vector representation, such as set inputs (Zaheer et al., 2017), it is not clear how the unimodal decoder can be applied.

### 2.2 HIERARCHICAL BAYES AUTOENCODER

We now present the Hierarchical Bayes Autoencoder (HBAE), which addresses the drawbacks of the VAE as discussed previously. The key insight is to design a multimodal decoder that can flexibily model the implicit (and explicit in the set case) one-to-many mapping from a latent code to target samples. To do so, we replace the decoder distribution of a VAE with an EBM-based decoder LeCun et al. (2006).

Formally, we start by using the joint distribution in Equation 1. We then define $p_\theta(\mathbf{x}|\mathbf{z}) = \frac{e^{-E(\mathbf{x}|\mathbf{z};\theta)}}{\int_{\mathbf{x}} e^{-E(\mathbf{x}|\mathbf{z};\theta)} d\mathbf{x}}$, with $E(\mathbf{x}|\mathbf{z}; \theta)$ being the energy function. From the graphical model perspective, this formulation removes the conditional independence assumption between the dimensions of $\mathbf{x}$ given $\mathbf{z}$, and allows for modeling rich interactions between them instead. This enables the decoder to assign high probability to multiple possible outputs, if needed (refer to Figure 2 for an illustration). Note that the EBM formulation of the decoder is a strict generalization of Gaussian and other exponential family energy-based models; see LeCun et al. (2006) for more explanations.

During training, we similarly apply a variational distribution $q_\phi(\mathbf{z}|\mathbf{x})$ as in a VAE, which leads to the following variational objective:

$$\min_{\theta,\phi} \mathrm{E}_{\mathbf{x} \sim p_{data}(\mathbf{x})}[-\mathrm{E}_{\mathbf{z} \sim q_\phi(\mathbf{z}|\mathbf{x})} \log p_\theta(\mathbf{x}|\mathbf{z}) + KL(q_\phi(\mathbf{z}|\mathbf{x})||p_\theta(\mathbf{z}))]$$

$$= \mathrm{E}_{\mathbf{x} \sim p_{data}(\mathbf{x})}[-\mathrm{E}_{\mathbf{z} \sim q_\phi(\mathbf{z}|\mathbf{x})}[\log \frac{e^{-E(\mathbf{x}|\mathbf{z};\theta)}}{\int_{\mathbf{x}} e^{-E(\mathbf{x}|\mathbf{z};\theta)} d\mathbf{x}}] + KL(q_\phi(\mathbf{z}|\mathbf{x})||p_\theta(\mathbf{z}))] \tag{2}$$

$$= \mathrm{E}_{\mathbf{x} \sim p_{data}(\mathbf{x})}[\mathrm{E}_{\mathbf{z} \sim q_\phi(\mathbf{z}|\mathbf{x})}[E(\mathbf{x}|\mathbf{z}; \theta) + \log \int_{\mathbf{x}} e^{-E(\mathbf{x}|\mathbf{z};\theta)} d\mathbf{x}] + KL(q_\phi(\mathbf{z}|\mathbf{x})||p_\theta(\mathbf{z}))],$$

where the first row is the standard variational lower bound of a VAE, and the following two rows are the result of plugging in the EBM defination of the decoder.

Directly minimizing Equation 2 as an objective function is difficult as it involves the logarithm of a partition function. As a solution, we turn to recent connections between deep EBMs and GANs (Zhao et al., 2016; LeCun et al., 2006; Wang & Liu, 2016; Zhai et al., 2016). As a brief recap, following Zhai et al. (2016), for an EBM with energy $E(\mathbf{x}; \theta)$, we can write its negative log likelihood

(NLL) as: $\mathrm{E}_{\mathbf{x} \sim p_{data}(\mathbf{x})}[E(\mathbf{x}; \theta)] + \log[\int_{\mathbf{x}} e^{-E(\mathbf{x}; \theta)} d\mathbf{x}]$, which can be further developed as:

$$\mathrm{E}_{\mathbf{x} \sim p_{data}(\mathbf{x})}[E(\mathbf{x}; \theta)] + \log \int_{\mathbf{x}} q_{\psi}(\mathbf{x}) \frac{e^{-E(\mathbf{x}; \theta)}}{q_{\psi}(\mathbf{x})} d\mathbf{x} = \mathrm{E}_{\mathbf{x} \sim p_{data}(\mathbf{x})}[E(\mathbf{x}; \theta)] + \log \mathrm{E}_{\mathbf{x} \sim q_{\psi}(\mathbf{x})}[\frac{e^{-E(\mathbf{x}; \theta)}}{q_{\psi}(\mathbf{x})}]$$

$$\geq \mathrm{E}_{\mathbf{x} \sim p_{data}(\mathbf{x})}[E(\mathbf{x}; \theta)] + \mathrm{E}_{\mathbf{x} \sim q_{\psi}(\mathbf{x})}[\log \frac{e^{-E(\mathbf{x}; \theta)}}{q_{\psi}(\mathbf{x})}] = \mathrm{E}_{\mathbf{x} \sim p_{data}(\mathbf{x})}[E(\mathbf{x}; \theta)] - \mathrm{E}_{\mathbf{x} \sim q_{\psi}(\mathbf{x})}[E(\mathbf{x}; \theta)] + H(q_{\psi}),$$

$$\tag{3}$$

where $q_{\psi}(\mathbf{x})$ is an auxiliary distribution with parameters $\psi$, with $H(q_{\psi})$ denoting its entropy. We then let $D(\mathbf{x}; \theta) = -E(\mathbf{x}; \theta)$ and $q_{\psi}(\mathbf{x}) = \int_{\mathbf{z}'} p_{\psi}(\mathbf{z}') \delta(G(\mathbf{z}'; \psi)) d\mathbf{z}'$ (i.e., an implicit distribution defined by a generator $G$ and a known prior $p_{\psi}(\mathbf{z}')$), which leads to an optimization procedure as follows:

$$\min_{\theta} \max_{\psi} \mathrm{E}_{\mathbf{x} \sim p_{data}(\mathbf{x})}[-D(\mathbf{x}; \theta)] + \mathrm{E}_{\mathbf{z} \sim p_{\psi}(\mathbf{z})}[D(G(\mathbf{z}; \psi); \theta)] + H(q_{\psi}), \tag{4}$$

In practice, we rely on implicit entropy regularizers such as batch normalization (Ioffe & Szegedy, 2015) in place of the entropy term $H(q_{\psi})$, which gives us a learning objective:

$$\min_{\theta} \max_{\psi} \mathrm{E}_{\mathbf{x} \sim p_{data}(\mathbf{x})}[-D(\mathbf{x}; \theta)] + \mathrm{E}_{\mathbf{z} \sim p_{\psi}(\mathbf{z})}[D(G(\mathbf{z}; \psi); \theta)], \tag{5}$$

which can be essentially implemented as a GAN, in particular a WGAN (Arjovsky et al., 2017), with $D$ being the discriminator and $G$ being the generator.

We then accordingly apply the same derivation and change of variables to Equation 2, and obtain a practical objective for HBAE:

$$\min_{\theta, \phi} \max_{\psi} \mathrm{E}_{\mathbf{x} \sim p_{data}(\mathbf{x})}[\mathrm{E}_{\mathbf{z} \sim q_{\phi}(\mathbf{z}|\mathbf{x})}[-D(\mathbf{x}|\mathbf{z}; \theta) + \mathrm{E}_{\mathbf{z}' \sim p_{\psi}(\mathbf{z}'|\mathbf{z})} D(G(\mathbf{z}'|\mathbf{z}; \psi)|\mathbf{z}; \theta)] + KL(q_{\phi}(\mathbf{z}|\mathbf{x})||p_{\theta}(\mathbf{z}))].$$

$$\tag{6}$$

Here both the discriminator $D$ and generator $G$ are conditional on $\mathbf{z}$, which is a sample from the encoder $q_{\phi}$. The additional noise distribution $p_{\psi}(\mathbf{z}'|\mathbf{z})$ can also be conditional, but we set it to a parameter free Gaussian distribution in practice. Furthermore, we can also extend the objective above in a similar way to other VAE variants such as Higgins et al. (2017); Zhao et al. (2017); Tolstikhin et al. (2017) where the KL divergence term is generalized to a weighted distance measure between the aggregated posterior distribution $q_{\phi}(\mathbf{z}) = \mathrm{E}_{\mathbf{x} \sim p_{data}(\mathbf{x})} q_{\phi}(\mathbf{z}|\mathbf{x})$ and the prior $p_{\theta}(\mathbf{z})$, which gives rise to our final objective:

$$\min_{\theta, \phi} \max_{\psi} \mathrm{E}_{\mathbf{x} \sim p_{data}(\mathbf{x})}[\mathrm{E}_{\mathbf{z} \sim q_{\phi}(\mathbf{z}|\mathbf{x})}[-D(\mathbf{x}|\mathbf{z}; \theta) + \mathrm{E}_{\mathbf{z}' \sim p_{\psi}(\mathbf{z}'|\mathbf{z})} D(G(\mathbf{z}'|\mathbf{z}; \psi)|\mathbf{z}; \theta)]] + \beta \mathcal{D}(q_{\phi}(\mathbf{z})||p_{\theta}(\mathbf{z})),$$

$$\tag{7}$$

where $\mathcal{D}$ denotes a distance between distributions, which we choose the MMD distance as in Tolstikhin et al. (2017), $\beta$ is a regularization factor.

The training procedure of HBAE is similar to that of a conditional GAN, except that the conditioning variable is learned jointly. To be concrete, given a fixed encoder $q_{\phi}$ and discriminator $D$, the conditional generator $G$ is updated in the inner loop to increase the conditional discriminator's score on $G$'s output. In the outer loop, the encoder $q_{\phi}$ and discriminator $D$ are jointly updated adversarially against $G$. During inference, we can sample from the generator as with other generative models. However, in a HBAE, sampling follows a two step hierarchy. One first sample a latent variable $\mathbf{z}$, either from the encoder distribution $q_{\phi}(\mathbf{z}|\mathbf{x})$, or from the prior $p_{\theta}(\mathbf{z})$. One then samples from a secondary $\mathbf{z}' \sim p_{\psi}(\mathbf{z}'|\mathbf{z})$ to generate an example. This sampling procedure resembles the inference process of a Hierarchical Bayes model, and hence we name it Hierarchical Bayes Autoencoder.

## 2.3 SET-HBAE

Given the HBAE formulation, it is also easy to extend the same generative modeling setting to cases where inputs are sets of images (e.g., different views of the same object). We refer to this variant as the Set-HBAE. In order to do so, we explicitly express the input in the set format $\mathbf{x} = \{\mathbf{x}^1, \mathbf{x}^2, ..., \mathbf{x}^k\}$ where $k$ denotes the size of the set. By construction, we let the elements within a set $\mathbf{x}$ to be conditionally independent given the set latent code $\mathbf{z}$ (see Figure 1). We then accordingly modify the HBAE objective Equation 7 as:

$$\min_{\theta, \phi} \max_{\psi} \mathrm{E}_{\mathbf{x} \sim p_{data}(\mathbf{x})}[\mathrm{E}_{\mathbf{z} \sim q_{\phi}(\mathbf{z}|\mathbf{x})}[-\frac{1}{|\mathbf{x}|} \sum_{\mathbf{x}^i \in \mathbf{x}} D(\mathbf{x}^i|\mathbf{z}; \theta) + \mathrm{E}_{\mathbf{z}' \sim p_{\psi}(\mathbf{z}'|\mathbf{z})} D(G(\mathbf{z}'|\mathbf{z}; \psi)|\mathbf{z}; \theta)]]$$

$$+ \beta \mathcal{D}(q_{\phi}(\mathbf{z})||p_{\theta}(\mathbf{z})), \tag{8}$$

where $|\mathbf{x}|$ denotes the size of the set. The encoder $q_\phi$ in this case takes form of a set encoder. Here we adopt the DeepSet (Zaheer et al., 2017) mechanism to average pool the codes of elements within the set; the form of the generator remains unchanged. During inference time, the hierarchical sampling procedure naturally corresponds to sampling a set code $\mathbf{z}$ first, followed by sampling an element code $\mathbf{z}'$. In this way, the Set-HBAE is able to perform both stochastic set reconstruction, and well as unconditional set generation.

## 3 RELATED WORK

**Autoencoding generative models**. The HBAE is similar to a handful of existing works which involve an encoder, decoder and a discriminator. Notably, BiGAN (Donahue et al., 2016) trains a two channel discriminator that tries to separate the two joint distributions $p(\mathbf{z})p(\mathbf{x}|\mathbf{z})$ and $p(\mathbf{x})p(\mathbf{z}|\mathbf{x})$, which involves an encoder and decoder, respectively. BiGAN differs from the HBAE in that the mapping between $\mathbf{z}$ and $\mathbf{x}$ is unimodal. Also, in a BiGAN, the encoder and decoder work jointly against the discriminator, whereas in the HBAE, the encoder and the discriminator work jointly against the generator. The VAE-GAN (Larsen et al., 2015) is a VAE/GAN hybrid model that consists of the three modules. In addition to the deocder being unimodal, a VAE-GAN deploys an unconditional discriminator, which contrasts with the conditional discriminator used in HBAE.

**Conditional GANs**. A conditional GAN (cGAN) (Mirza & Osindero, 2014; Miyato & Koyama, 2018) modifies the generator and discriminator of a GAN to be conditional on class labels. The HBAE can be considered a generalization of cGANs, as it learns the conditioning representation jointly with the generator and discriminator. To this end, it is possible to come up with a semi-supervised learning version of HBAE where a small portion of input data have observed labels, which are jointly trained with unlabelled data. Refer to Figure 2 for a comparison of the HBAE with the cGAN, BiGAN and VAE-GAN architectures.

**PixelCNN-based multimodal decoders**. Another line of work that employs a multimodal encoder is represented by PixelVAE (Gulrajani et al., 2016), which uses a PixelCNN-based decoder (Van den Oord et al., 2016). While similar in spirit to our work, PixelCNN suffers from capturing long term dependency of images, and is also expensive during inference time. We see HBAE as providing an alternative approach to this line of work, which leverages the power of state of the art GANs for decoding.

**VAEs with large latent codes**. Another design philosophy of VAEs is represented by VQ-VAE2 (Razavi et al., 2019), where a large latent code is used for each example, with preserved spatial dimensions. VQ-VAE2 then learns a powerful prior in the form of a PixelCNN, which is able to generate good quality samples at inference time. We note that in this scenario, the use of a unimodal decoder is well justified where the latent space is large enough to allow learning a one-to-one mapping between a latent code and an input. The HBAE on the other hand, works best in the setup where a compact latent code is used, or in applications where explicit multimodality is desired, such as with sets as inputs.

**Generative Modeling of sets**. Generative modeling of sets is a challenging task. Hierachical Bayes models have previously been widely applied to document modeling, which can be considered as a set of words, represented by the LDA model (Blei et al., 2003). However, due to the discrete nature of words, LDA only needs to rely on a hierarchy of simple distributions such as the Dirichlet and multinomial distributions. Recently, there has been a a body of work dedicated to point cloud generative modeling, where sets are composed of 3D points in Euclidean space (Li et al., 2018; Achlioptas et al., 2017). However, sets considered in our work contain much richer structure than point clouds, which contain very little structure per element. It is thus unclear how methods developed from the point cloud or document modeling lines of work generalize to more complex data.

## 4 EXPERIMENTS

### 4.1 ARCHITECTURE

The HBAE defined in Equation 7 has three trainable modules: an encoder which corresponds to $q_\phi(\mathbf{z}|\mathbf{x})$[1], a conditional discriminator $D(\mathbf{x}|\mathbf{z};\theta)$, and a conditional generator $G(\mathbf{z}'|\mathbf{z};\psi)$. Our architecture design largely mimics that of a conditional GAN, in particular the projection-based discriminator proposed in Miyato & Koyama (2018). Here the latent code, $z$, takes place of the label, $y$, in a conditional GAN, and is jointly learned with other modules instead of being given. We follow a simple design choice for the architectures, where both the encoder and discriminator consist of stacks of residual blocks with Spectral Normalization (Miyato et al., 2018), and the generator consists of stacks of residual blocks with Batch Normalization (Ioffe & Szegedy, 2015). Refer to Figure 3 for more details.

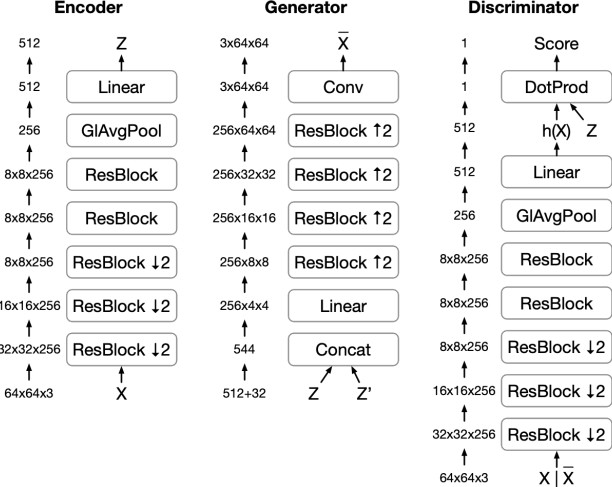

Figure 3: HBAE implementation. Left: Encoder. Middle: Conditional generator. Right: Conditional discriminator, which takes as input ground truth $\mathbf{x}$ or sample $\bar{\mathbf{x}}$. The discriminator score is a dot product of the intermediate representation and the latent code $D(\mathbf{x}|\mathbf{z}) = h(\mathbf{x})^T \mathbf{z}$.

### 4.2 TRAINING PROTOCOL

In all the experiments, we follow the training protocols of standard GANs as in Donahue et al. (2016), where the encoder and discriminator are trained jointly against the generator, as shown in Equation 7. We use Adam with a learning rate 2e-4 for the generator , and learning rate 4e-4 for the encoder and discriminator. The moment terms are set as $\{\}$ for both optimizers. In practice, one failure mode we have found by directly following Equation 7 is that the encoder can collapse to a constant mapping which makes HBVE regress to an unconditional GAN. As a remedy, we put an additional reconstruction regularization term on the encoder's parameters $R(\phi) = \lambda \mathrm{E}_{\mathbf{x} \sim p_{data}(\mathbf{x})} \mathrm{E}_{\mathbf{z} \sim q_\phi(\mathbf{z}|\mathbf{x})} \|\mathbf{x} - G(\mathbf{0}|\mathbf{z})\|_1$ (for the Set-HBAE, this corresponds to mapping $\mathbf{z}$ to all elements within the set: $\lambda \mathrm{E}_{\mathbf{x} \sim p_{data}(\mathbf{x})} \mathrm{E}_{\mathbf{z} \sim q_\phi(\mathbf{z}|\mathbf{x})} \frac{1}{|\mathbf{x}|} \sum_{\mathbf{x}^i \in \mathbf{x}} \|\mathbf{x}^i - G(\mathbf{0}|\mathbf{z})\|_1$). Here $\lambda$ is the regularization strength which we set as $1$. We note that this regularization term is not essential for HBAE to train, but in most cases helps with the stability.

### 4.3 SINGLE IMAGE INPUTS

In the first set of experiments, we test the HBAE in the standard setting where inputs are single images. We use the CelebA Dataset (Liu et al., 2015) with resolution $64 \times 64$. In Figure 4, we show some of the qualitative results. We see that the HBAE is able to reconstruct input examples in a

---

[1]In our implementations, we adopt a deterministic encoder which corresponds to setting $q_\phi$ as a $\delta$ distribution, which gives the most stable results.

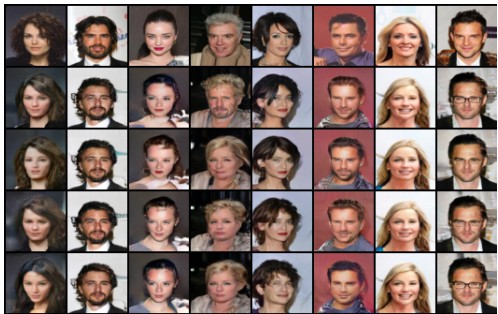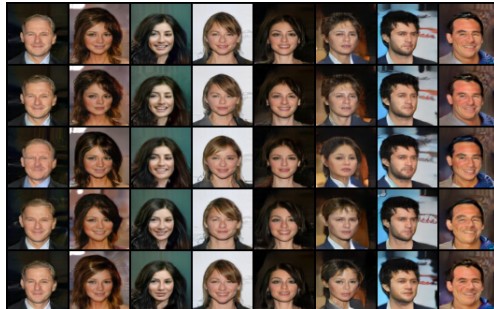

Figure 4: Reconstructions and samples from HBAE. Left: input images from the test set (first row) and four stochastic reconstructions with different $\mathbf{z}'$ (second through last rows). Right: unconditional samples (first row) and four stochastic samples with shared $\mathbf{z}$ and different $\mathbf{z}'$.

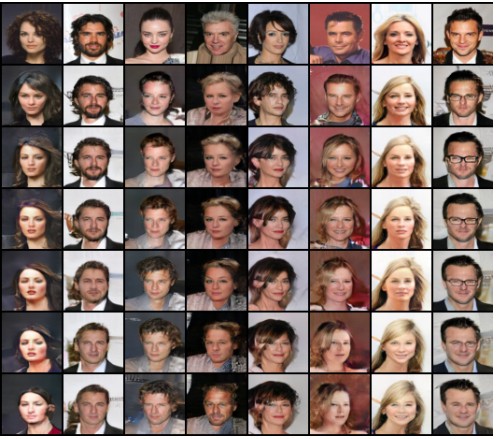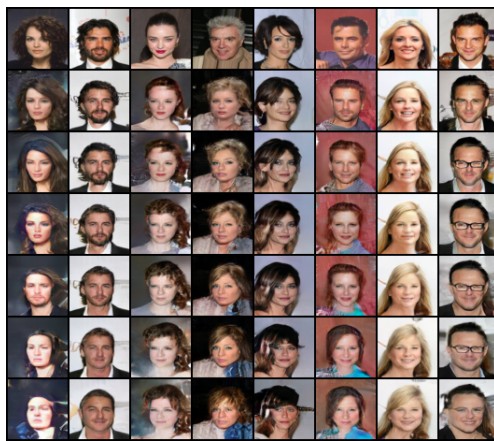

Figure 5: Markov chains starting from real examples from the test set (top row) and 6 recursive consecutive stochastic samples (second through last rows). We run two consecutive Markov chains staring from the same examples, corresponding to columns in the left and right plots.

semantically meaningful way by preserving certain aspects of the input face, while outputting variations that exhibit rich and changing details, such as adding or removing glasses, varying expressions subtly, or changing hairstyles. When sampled unconditionally, the HBAE is also able to produce sharp examples. As control experiments, we have accordingly trained a GAN and VAE using the same architectures (encoder and generator), and reported the FID score of the unconditional generations, and reconstructions (when applicable), in Table 1. We have also included the reconstruction MSE of the VAE and HBAE evaluated in the pixel space and Inception V3's feature space. Refer to the caption for detailed discussions.

Additionally, we show that the HBAE can be used to sample images in a Markov chain, exploring the data manifold around an example as a random walk. The Markov chain starts from a real example, and recursively passes it through the encoder and the generator. We show two such chains in Figure 5, starting from the same test examples, both of which exhibit smooth transitions from one sample to the next, without burning into a single mode. Interestingly, the two chains also lead to different sampling paths, due to the multimodality of the generator. This behavior is not possible with regular VAE or GAN formulations.

## 4.4 SET OF IMAGES AS INPUT

In the second set of experiments, we change the inputs to be a set of images. The first dataset we use is the ShapeNet dataset from Chang et al. (2015), which consists of 3d objects with projected 2d views. We use a subset which contains 13 popular categories, namely *airplane, bench, cabinet, car, chair, monitor, lamp, speaker, gun, couch, table, phone, ship*. The training set consists of 32,837

Table 1: FID scores together with reconstruction MSE evaluated in pixel space and Inception's feature space. In a control experiment setting, HBAE achieves significantly better FID scores than the VAE counterpart, while being more comparable to unconditionally trained GANs. As a reference, the FID score for ground truth test data is 4.08. Next, reconstruction MSE for the HBAE is significantly larger than the VAE counterpart evaluated in pixel space, as expected given that the decoder is multimodal and stochastic. However, when evaluated in Inception's feature space, HBAE and VAE metrics are comparable. As a reference, the Inception MSE for a random pair of test images is 0.126.

| Model | FID recon | FID gen | MSE Inception | MSE pixel |
|---|---|---|---|---|
| HBAE (ours) | 12.08 | 16.72 | 0.076 | 0.105 |
| VAE | 44.57 | 65.97 | 0.071 | 0.019 |
| GAN | - | 9.13 | - | - |

Table 2: Few shot classification results on ShapeNet (Chang et al., 2015). We show the test accuracy of each features while varying the number of training examples. Here random denotes uninitialized features with the same architecture, and supervised denotes the supervised training setting with the same encoder architecture.

| # Training example | 64 | 128 | 256 | 512 | 1024 | 2048 |
|---|---|---|---|---|---|---|
| Set-HBAE (ours) | **50.5** | **56.0** | **65.1** | **70.8** | **74.0** | 76.2 |
| VAE | 39.2 | 49.9 | 56.3 | 60.0 | 60.7 | 62.9 |
| Random | 19.2 | 19.2 | 19.2 | 19.2 | 19.2 | 19.2 |
| Supervised | 44.7 | 52.2 | 60.5 | 64.4 | 70.2 | **76.3** |

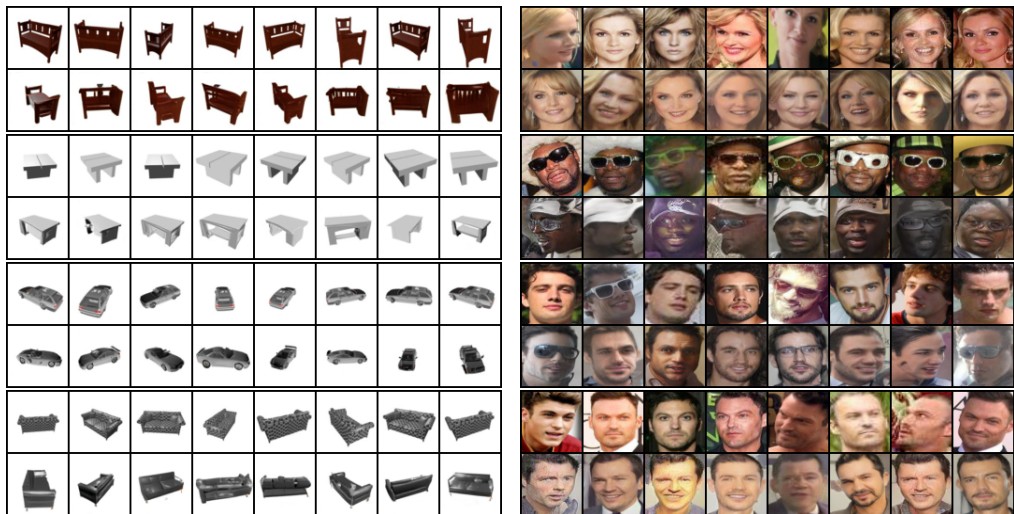

Figure 6: Reconstructions of the ShapeNet (left figure) and VGGFace2 (right figure) daatasets with a Set-HBAE. For each block, the first row corresponds to an input set from the test split, and the second row shows one (stochastic) reconstruction of the set from the Set-HBAE.

unique objects and 788,088 images in total. The second dataset we use is the VGGFace2 dataset from Cao et al. (2018), which is a face dataset containing 9,131 subjects and 3.31M images.

For both datasets, we randomly construct fixed-size sets of 8 elements. For ShapeNet, this is done by randomly selecting an object and 8 of its views; for VGGFace2, we similarly select one identity and 8 of her/his faces. A mini-batch is constructed by selecting $N$ (which we set to 32) objects/subjects, which amounts to $Nx8$ images. Example set inputs together with their stochastic reconstructions are shown in Figure 6. The qualitative results indicate that the samples are diverse, yet are consistent with each other and tend to resemble the semantic traits of their respective input sets (i.e., same object in different views for the ShapeNet case and same identity with varying attributes for the VGGFace2 case).

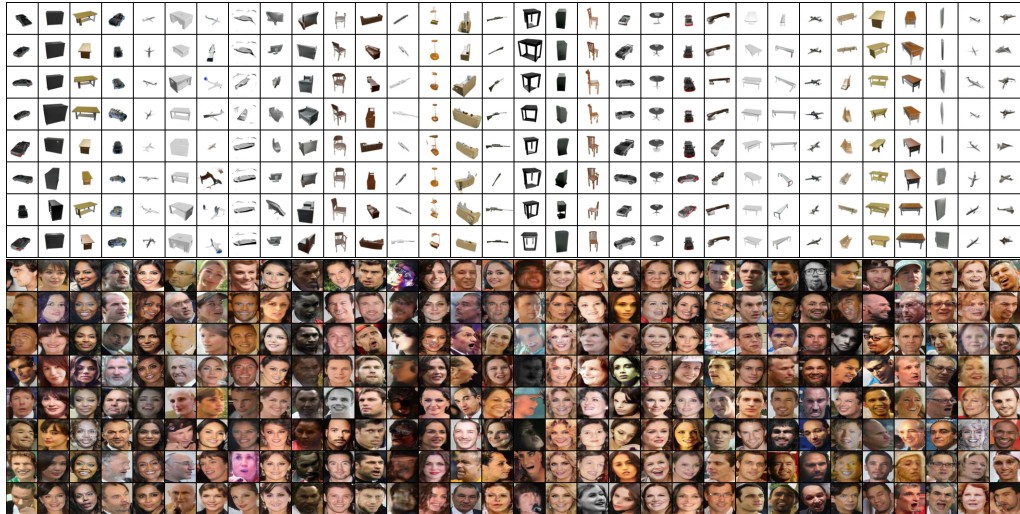

Figure 7: Uncurated unconditional generations from a Set-HBAE trained on ShapeNet (top) and VGGFace2 (bottom) with set size fixed to 8. Each column corresponds to a sampled set. The Set-HBAE generates sets that are coherent, and often with good visual quality.

As a further quantitative measure, we also conducted few shot classification experiments on ShapeNet with the learned features. We compare the features learned from our Set-HBAE with a control VAE, as well as with randomly initialized features, and with supervised training results, all with the same encoder architecture. For all settings, we trained a linear classifier with crossentropy loss, without any explicit data augmentation or regularization (except for the supervised learning setting where the encoder is jointly trained with the linear classifier). We vary the number of training images from 64 to 2048, and report the results in Table 2. We see that in the few shot classification domain, the learned features derived from the Set-HBAE largely outperform the supervised learning approach as well as the VAE results, all of which far exceed random features.

Finally, we show the Set-HBAE's ability to generate unconditional samples. The generated results are shown in Figure 7 for both ShapeNet and VGGFace2. Notably, the Set-HBAE is able to generate diverse sets, where elements within the generated sets are also diverse but consistent in ways that preserves object shape or face attributes. To the best of our knowledge, this is the first generative model that is able to generate complex sets with richly varying yet coherent structure.

## 5 CONCLUSION

In this paper, we presented the HBAE, a novel probabilistic generative model with an EBM-based multimodal decoder. In our implementation, we jointly trained the encoder with variational inference and the EBM decoder with a powerful conditional GAN. We demonstrated that the decoder is capable of generating diverse conditional and unconditional samples. Then we introduced the Set-HBAE, a novel generative model for sets of examples, generalizing the HBAE framework. The Set-HBAE takes sets as input and can sample plausible variations that are consistent with the conditioning set, but generalize beyond the specific input instances. In addition, we show that the

Set-HBAE performs well on a ShapeNet Chang et al. (2015) few-shot classification task, showing the advantage of leveraging generative models in the small data regime. The ability to sample plausible variations conditioned on an input example or set of examples is a key building block for enabling predictive models, for which we see future applications in areas such as video (Lee et al., 2018), 3D scene generation (Eslami et al., 2018), and model-based reinforcement learning (Ha & Schmidhuber, 2018).

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

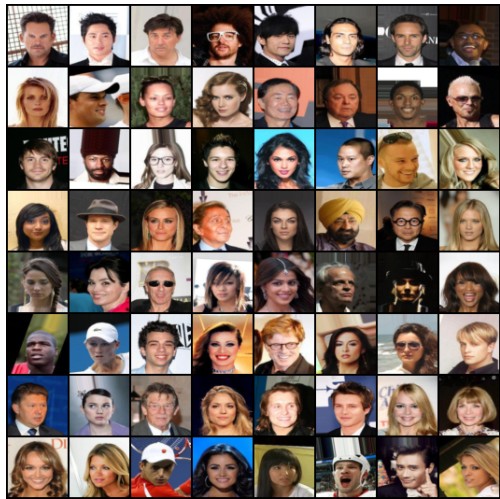
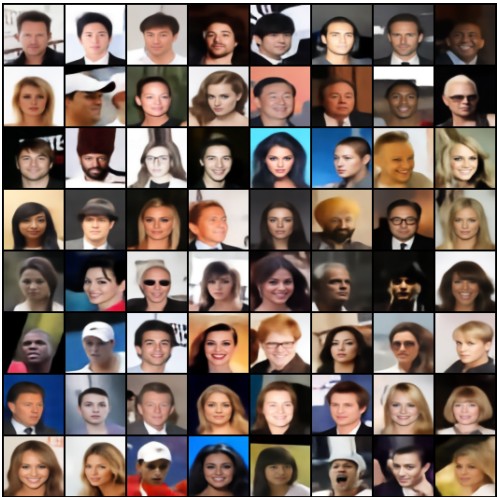

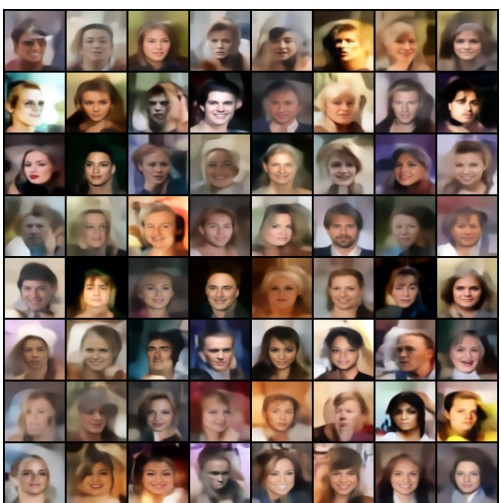

Figure 8: VAE results on the Celeba Dataset. Top left: random samples of Celeba image; top right: reconstructions of a VAE; bottom: unconditional samples from a VAE.

# A    APPENDIX

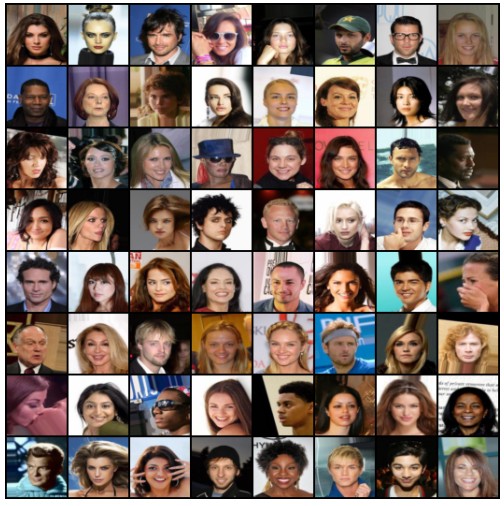 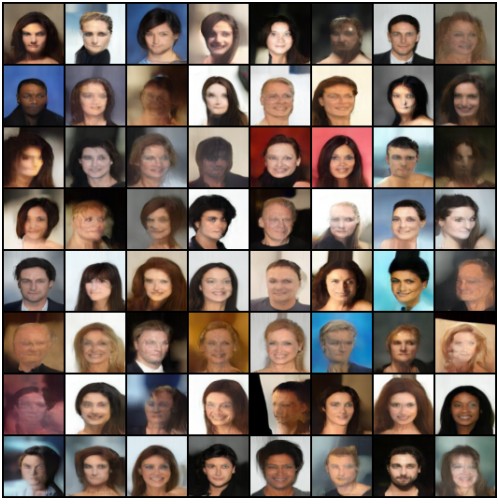

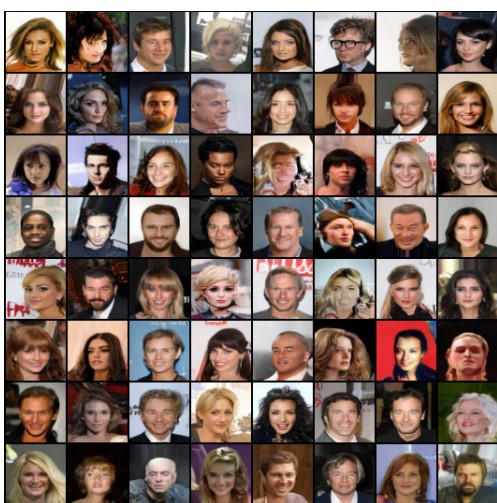

Figure 9: Unconditional GAN results on the Celeba Dataset. Top left: random samples of Celeba image; top right: reconstructions of a GAN; bottom: unconditional samples from a GAN. The reconstructions are achieved by first training the unconditional GAN then train an additional encoder with the generator as decoder with $\ell_1$ loss.

