# OpenReview forum: "Hierarchical Bayes Autoencoders"
_ICLR.cc/2020/Conference — Reject_

### Official Review · AnonReviewer3 · 2019-10-09
**Official Blind Review #3**

**Rating:** 1

**Review:**

The article proposes to replace the conditional distribution p(x|z) in an VAE by a general graphical model parameterized by a deep neural network. Using a series of approximations it arrives at a triplet of models, where the graphical model is split into a model representing the likelihood and a model that generates samples from the distribution. the third model is the usual autoencoder. The core idea is that the latent space of the VAE serves as bottleneck so that the graphical model develops a multi-modal distribution. At the same time, the use of a neural network allows for a much better modeling than using the typical local Gaussian distribution as is used in most VAE-approaches. Experimental results show that reconstructions and markov-chains starting from reconstructions produce decent results. An extension to sets is proposed as well as a task showing performance for semi-supervised learning.

-------------------------------------------
The text above serves mainly as a  summary of what the paper sets out to do. However, we have to check whether the paper is actually achieving this goal. In my view, it does not, and it suffers from severe problems on the theoretical side. Furthermore, important evaluations that are standard within the VAE framework are missing: samples from the true underlying distribution. In the remainder of this review, I will try to substantiate the claims regarding theory which lead me to an overall decision to reject the article in its current form.

Let us start from a birds-eye perspective: the model consists of three parts:
the Encoder E, which i will refer to as q(z|x)
the Energy-function, also called E(x|z) and with -E referred to as D later on.
a generator function G(z,z') that is supposed to sample from the distribution p(x|z) = 1/Z(z) exp(-E(x|z)) using entropy generated by z'~N(0,I). I will refer to this distribution as r(x|z)

This split is smart: using a general graphical model to model p(x|z) makes it very hard to sample from the distribution and it is difficult to estimate the normalization constant Z. The paper refers to Zhai et.al (2016) to find eq(3). I really like (3) because the inequality is tight when G is producing samples from p(x|z). To be more exact, we have that the estimator for the normalization constant is log(Z) - KL(r(x|z) || p(x|z) ). This reveals the first issue: the generator must have an r(x|z) > 0 for all x except a countable infinite subset. This is difficult to achieve with dimensionality of z and z' smaller than dimensionality of x and G being deterministic. Therefore, KL(r(x|z) || p(x|z) ) is already infinite. This is a technicality that i will ignore, since we can always add a minimal amount of noise on the output of G, even though this paper does not.

The real problem starts with eq(4). Since r(x|z) is implicit, there is no way to compute its entropy, which is then just discarded. The paper argues that this is no problem, because they intend to somehow bound the entropy. We will look at this point, later.

The issue with bounding is that for this to be meaningful, we also need a bound on the entropy of p(x|z), because otherwise there is no way for r(x|z) to achieve a small KL-divergence. If we now look at (4), we can compute the optimal r without bounded entropy. The optimal r is the distribution that by (5) maximizes -E(x|z) for a given z. this result is not a multi-modal distribution but a delta-peak on the optimum and certainly not p(x|z). so the bound is super important and technically we need a bound on log|det(d/dz' G(z,z'))|.

This is difficult from a theoretical view-point as well as from a modeling view-point because we know that we have to have nearly discontinuous transformations to create sharp edges. Further, to transform z'~N(0,I), into a multi-modal distribution, there must be a function g(z')->w so that w is multi-modal. This is difficult to achieve with bounded determinant.

Looking at the experiments, let's see whether the determinant is actually bounded. The paper claims that bounding can be achieved using batch-normalization. First this requires a reference on why this should be the case and show that this holds as implemented for the resnet block and second, this requires an estimate for how large the entropy can still be. if the entropy can be very large, we can not claim to train a VAE, because the error going from (3) to (4) is large. Secondly, we have to ensure that both E and G have bounded entropy, because otherwise KL(r(x|z) || p(x|z) ) will be large and there is no way to that the approximation in (3) is good, at which point E can not be claimed as doing maximum-likelihood on the data.

As shown in Figure 3, the Generator has a Conv-Block and the initial layer is a linear layer. Similarly the discriminator (or E) has a linear output layer. Therefore, the entropy is not bounded as the paper does not claim to have any entropy normalization here. The paper does not say what the activation functions are, but if the final layer of E is linear or Relu, we can add yet another problem to the list (next to the unbounded Jacobian): the VAE framework stops making sense. If we look at eq (6) or (7), the final divergence term (multiplied by beta) hinges on the fact that it provides a suitable error contribution. However, if the final layer of D is linear, we can just scale that layer arbitrarily large until the error contribution of the divergence vanishes.

Lastly, the Energy-function does not even necessarily model a probability function. Remember, we need that Z(z)=\int exp(-E(x|z)) dx <\infty. However, the definition by Figure (3) is E(x|z)=-h(x)^T(z) and the output layer of h is linear. so we can find weights such that h(x)=c \forall x. In which case
Z(z)=\int exp(-c^Tz) dx = exp(-c^Tz) \int 1 dx
which is only bounded if x is bounded, which I can not see since the generator is not bounded as well. This is only one of many counter-examples.


A few other small things:
- For z from a normal distribution the bottleneck discussion in 2.1 last paragraph does not apply, because for a dataset with N samples it is always possible to partition the real-space of Z into N partitions with probability 1/N each. This even works with 1-dimensional normal distributions, independently of the dimensionality of x. So a multi-modal distribution for p(x|z) is really not necessary if encoder and decoder are strong enough to memorize the dataset.
- below (5) it is claimed that this is similar to a WGAN, but WGAN have a difficult constraint to be 1-Lipschitz to make it work (which proves to be extremely difficult in practice as it translates to a similar inequality constraint on the jacobian)
- In Fig(5) it is unclear to me how a markov-chain is defined using the model. I assume this is done by taking samples from z', generating a new x and encoding this to obtain a new z. If z' models different features from z (assuming z' is multi-modal) we would assume that the z would remain the same in each point, so no exploration of p(z) takes place. However, we can already see in the lat rows that the pictures look more strange, so some mixing is taking place but it is not clear how that relates to p(z). Therefore i would really like to see images generated by sampling z~N(0,I) and z'~N(0,I) to see whether the generated images look reasonable.
- Moreover, to show that the Generator actually produces images that are likely under p(x|z), we would also need samples generated from the true underlying distribution p(x,z), e.g. using hamilton monte-carlo.
- Finally, i have to notice that the model, while called hierarchival VAE, does not provide a way to compute z' from the input. is that than really an autoencoder?

**Experience Assessment:**

I have published one or two papers in this area.

**Review Assessment: Checking Correctness Of Derivations And Theory:**

I carefully checked the derivations and theory.

**Review Assessment: Checking Correctness Of Experiments:**

I assessed the sensibility of the experiments.

**Review Assessment: Thoroughness In Paper Reading:**

I read the paper at least twice and used my best judgement in assessing the paper.

---

> ### Author Response · Authors · 2019-11-09
> **Author's response to Reviewer #3 part 1/2**
>
> We thank R3 for the technical critique. We address issues raised in each paragraph (referred to as P#num) below the dashed line. For all cases we will update the draft accordingly. R3’s main concern is around our approximation to the EBM; however, we’d like to highlight that our main contribution is a novel generalization of VAE’s decoder distribution (by using EBM as a tool), with experimental results supporting its effectiveness, including showing an application to generative modeling of sets of images for the first time.
>
> P1: “The text above …… in its current form.”
> We assume that by “samples from the true distribution” R3 means sampling from the prior distribution p(z). We have shown qualitative results in Figure(4) right panel and in Figure 7 for single image and image set settings, respectively. Quantitative results are also included in Table 1.
>
> P2-3: “Let us start from a birds-eye perspective …… even though this paper does not.”
> We agree that, in theory, the support of r(x|z) needs to be large enough to make sure that the KL term makes sense and then r(x|z) can possibly approximate p(x|z). This problem is well known in the GAN literature, where tricks like instance noise [1] were proposed to address this. Another way of alleviating this problem is by constraining p(x|z) (D in GAN’s context), as suggested by WGAN, which is what we adopt (by imposing Spectral normalization on every layer of D/E). Empirically, we have found this to work better than using instance noise.
>
> P4-5: “The real problem starts with eq(4) …… we need a bound on log|det(d/dz' G(z,z'))|.”
> Directly approximating the entropy of r(x|z) is difficult, thus as a result we do not directly attempt to do so. And yes in theory, optimizing equation (5) in its non-parametric form will lead to extreme mode collapse of G. We first apologize for not discussing this in full detail, which we will do in the updated draft, and then provide a justification for our implementation. For equation (5), our implementation approximates the max and min procedure with a few steps of mini-batch update, w.r.t. to D and G, respectively. When updating G for one step, with D being fixed, the parameter update essentially updates the generated examples from x -> x’, where x = G(z, \psi), x’ = G(z, \psi’), with \psi’ being the new parameters of G. With proper conditioning (e.g., with BN on G) and learning rate setup, one can control the update such that \|x’ - x\| remains a small quantity, meaning that x’ is a local neighborhood of x that decreases the energy. This is very much akin to what one will do for one step stochastic MCMC (see [2] for its recent application in deep EBMs), minus the noise to the gradient. As a result, a properly controlled G update approximates stochastic MCMC, which is an unbiased estimator of the unnormalized density. Approximations to MCMC have seen wide applications in the EBM literature, such as contrastive divergence [3]. Of course, MCMC itself suffers from problems of not covering enough modes of the true density (entropy lower than the true model), but it has served as a valuable tool in the bayesian inference community and has been proven useful in many applications. As a result, G will suffer from the same problem to some degree, but can still provide a reasonable approximation to the true density. A side note is that for our implementations log|det(d/dz' G(z,z’))| usually doesn’t exist, as z’ is usually not of the same dimensionality as x.
>
> P6: “This is difficult from a theoretical view-point ……This is difficult to achieve with bounded determinant”
> We do not agree that our formulation of G suffers from capacity issues of learning a multimodal mapping. This is best shown in flow based models, see [4] for such an example, where z’ by construction has the same dimensionality as x, the log determinant thus exists and can be bounded with proper implementations. The success of flow based models on real world dataset suggests that modern neural networks do have the capacity of constructing such a multimodal mapping from a normal distribution. The same argument can be applied to a generator of a regular GAN, where the determinant also does not exist, but it has been shown to be able to achieve the multimodal mapping from either normal or uniform distributions to real images. Our experiment results also show that our stochastic reconstructions are indeed multi-modal (figure 4, left panel).

---

> > ### Author Response · Authors · 2019-11-09
> > **Author's response to Reviewer #3 part 2/2**
> >
> > P7: “Looking at the experiments …… maximum-likelihood on the data.”
> > The core of the question is whether it is possible to train a reasonable EBM without an estimate or control over KL(r(x)||p(x)), with r(x) being any distribution that approximates p(x). We believe the answer is yes, given all the successful empirical applications of EBMs (Restricted Boltzmann Machines as a special case) trained with MCMC variants (Contrastive Divergence for example), also a recent contribution [6] that is extremely similar to our treatment where a generator is used in place of MCMC. In our experiments we also see that our HBAEs are able to learn to produce reasonable reconstructions. But we also argue that fully answering this question is beyond the scope of this work, because exactly the same question can be asked to the larger approximate inference community, e.g., how to tell and quantify that a MCMC chain has mixed and to what extent it affects the learning algorithm that relies on the samples produced.
> >
> > P8: “As shown in Figure 3 …… contribution of the divergence vanishes”
> > We apologize for omitting some of the implementation details, which we will update later in the draft. The generator has a tanh activation in the end, so it’s outputs are in the space of [-1, 1]^d where d is the dimensionality of an image (h*w*3). Thus the entropy of the generator is upper bounded by that of the uniform distribution on the same space. Similarly the entropy of p(x|z) is also upper bounded by the same quantity. But again, both are difficult to quantify. Regarding the scaling issue of D, the scale of D is upper bounded because of the use of Spectral Normalization on every layer (including the last linear layer, which does not have non-linearity). Also another thing we have mentioned but did not state explicitly is that we use the hinge loss version of Equation (5), instead of it’s raw form, similar to modern GAN implementations such as found in BigGAN [5], which further prevents the arbitrary growth of D’s scale. In practice, we have found the two tricks sufficiently stabilizes the contribution of the divergence term.
> >
> > Pou9: “Lastly, the Energy-function does not even …… many counter-examples.”
> > Again, x is bounded as stated above, in the range [-1, 1]^d. Secondly, as a minor note, Z(z)=\int exp(-E(x|z)) dx <\infty is not a necessary condition to ensure that the EBM is valid. Rather instead, you only need to have Z(z)=\int exp(-E(x|z) + min_x{E(x|z)}) dx <\infty, which is clearly bounded when x is in range [-1, 1]^d.
> >
> > A few other small things:
> > “For z from a normal distribution …… to memorize the dataset.”
> > This is an interesting argument. The VAE is a regularized auto encoder, which essentially prevents extreme memorization of examples by imposing an information bottleneck regularization on the representation, in the form of KL(q(z|x)||p(z)) with p(z) being a simple prior (a prior with relatively high entropy). The example given by R3 thus will incur a high KL term loss and will likely not learn any useful information like what a VAE does. Plus, a VAE will work perfectly with N -> infinity given that all examples are drawn from the same distribution, while the counterexample will not hold.
> >
> > “below (5) it is claimed that this is similar to a WGAN …… inequality constraint on the jacobian”
> > We agree that Equation (5) will suffer from the same difficulty of training as a WGAN, and we rely on Spectral Normalization instead, which works well in practice (see [5]).
> >
> > “In Fig(5) it is unclear to me how …… images look reasonable.”
> > Yes R3’s understanding of Fig(5) is correct, will clarify. However, we do show samples from p(z) already in Fig(4) (right panel), with varied z’, which seems to be what R3 is requesting.
> >
> > “Moreover, to show that the Generator …… using hamilton monte-carlo”
> > This a good suggestion, we will add this experiment if time permits.
> >
> > “Finally, i have to notice that the model …… is that than really an autoencoder?”
> > HBAE is a stochastic autoencoder, where there are multiple reconstructions instead of one. Note that this is the same as VAE, in which the “decoder” is a Gaussian distribution, which is also stochastic.
> >
> > Refs:
> > [1] Amortised MAP Inference for Image Super-resolution, Sønderby et al
> > [2] Implicit Generation and Generalization in Energy-Based Models, Du and Mordatch
> > [3] On Contrastive Divergence Learning, Carreira-Perpin ̃ ́an and Hinton
> > [4] Glow: Generative Flow with Invertible 1x1 Convolutions, Kingma and Dhariwal
> > [5] Large Scale GAN Training for High Fidelity Natural Image Synthesis, Brock et al
> > [6] Maximum Entropy Generators for Energy-Based Models, Kumar et al

---

### Official Review · AnonReviewer1 · 2019-10-22
**Official Blind Review #1**

**Rating:** 3

**Review:**

In this work a novel probabilistic generative model is introduced mixing several existing frameworks: The Hierarchical Bayes Autoencoder (HBAE) can be interpreted as a cGAN with a VAE encoder. They also incorporate a flexible learned multimodal decoder in the form of an EBM. The authors claim to produce stochastic reconstructions varying around the local data manifold of examples, and diverse unconditional samples. In addition, they present an extension of their HBAE formulation in order to model sets of inputs. This is one of the main contributions of this work since generative modeling of sets is a challenging and unsolved task.

Although there is a clear explanation of the contributions of this paper, the motivation of this study is not precisely described. The introduction provides a good understanding of the topic; however, the authors may wish to provide several examples on the interest of the study. They provide a good description of the existing generative modeling architectures and the applications involved but not mention the importance of their contributions to the real world.

The proposed formulation seems encouraging thanks to the incorporation of multimodal decoders. The derivation of HBAE is theoretically well justified, as a result, one could replicate or further work in this paper. The authors explain in detail the procedure they took in order to arrive to the final HBAE formulation giving a clear understanding of the topic they present.

By describing the related works, they give an understandable perspective about the different drawbacks and differences of the existing methods. One could deduce the motivations of this work thanks to this section, however they should have been state clearer at the beginning.

The authors explain in detail all the different parts of the architecture, presenting precise information about the different layers. At first sight, it seems like one could reproduce the methodology used. It is clearly explained and discussed.

I think the experiments conducted to evaluate this work are not enough. The authors performed several experiments in order to prove the efficiency of their methodology by showing the generated images and investigate the qualitative results. If we look at the results in figure 4, we can observe some artifacts and blur regions in the image. Moreover, the variations on the faces sometimes seem more deformations than a different human feature. It would have been interesting to evaluate the identity preservation to further investigate the quality of the images. Although the methodology, conceptually, suggested an interesting approach, the results are not so encouraging given the quality of the images achieved by other methodologies.

Apart from comparing quantitatively the capacity of the generated images with other related works, it would have been interesting to show that qualitatively by showing images from the mention methods. In general, there is some information missing in order to replicate the results and more details would have been appreciated.

Regarding the experiments about the set of images as input, the quality of the images is similar to the simple HBAE. Since this is a more challenging approach and one could not compare with so many existing methodologies, it seems like they are going to the right direction to achieve the desired images. However, looking at the exposed results, the quality of the images is not realistic and has a lot of artifacts. Some other experiments could have been performed in order to show more interesting results.

Questions:

-	Could you better explain the meaning of figure 1? I think it was a good idea to exemplify the differences but is not clear enough.
-	Did you perform any face alignment when dealing with the different faces databases?
-	Did you think about the possibility of giving a concrete condition for varying the image?
-	Which could be some concrete applications for your work?


**Experience Assessment:**

I have published one or two papers in this area.

**Review Assessment: Checking Correctness Of Derivations And Theory:**

I assessed the sensibility of the derivations and theory.

**Review Assessment: Checking Correctness Of Experiments:**

I carefully checked the experiments.

**Review Assessment: Thoroughness In Paper Reading:**

I read the paper at least twice and used my best judgement in assessing the paper.

---

> ### Author Response · Authors · 2019-11-12
> **Author's response to Reviewer #1 part 1/2**
>
> We thank reviewer 1 for the comments. We are glad to see that the reviewer has found our theoretical contribution interesting. We would also like to emphasize that it is not our goal in this paper to achieve the state-of-the-art results on generative modeling of images, but rather we focus on showing a new framework that enables solving novel and challenging problems (e.g., generating self consistent sets of images), which we will further elaborate below.
>
> 1. Motivation and application of HBAE
> We apologize for not properly motivating HBAE and agree that part of the materials in the related work could be adopted to the introduction section, which we will do. Here we provide two additional points w.r.t. the potential application of HBAE. 1) Having the ability to stochastically reconstruct inputs in a generative model has important applications in video prediction [1] and model based reinforcement learning [2]. In particular, both [1] and [2] utilize VAE as the building block for their respective applications, and both can be potentially improved with HBAE which produces much better reconstructions and generations compared with a standard VAE. Although we do not directly evaluate HBAE’s usefulness in the aforementioned applications, as it is outside the scope of the intended contribution, we believe such an extension is straightforward and we leave it as future work. 2) Generative modeling on sets of structured data with Set-HBAE, which fills in the gap between generative modeling of vectorized inputs and discriminative modeling of set inputs (e.g., see [4]). Given the success of applying generative modeling to discriminative learning on vectorized inputs (see, e.g., [5, 6]), it is reasonable to assume set generative models like Set-HBAE will play a similar role in set discriminative tasks like those in [4].
>
> 2. Absolute visually quality of generated examples in single image HBAE
> We agree that absolute visual quality is not as good as state-of-the-art GAN generation methods. However, we’d like to emphasize two points here. First, keep in mind that the ground truth images are at a low resolution of 64x64 for all experiments and bilinear interpolation is used for display; high-resolution modeling is out of the scope of the current work due to compute issues. Second, compared with VAE counterparts, our HBAE outputs drastically better (and stochastic) reconstructions and unconditional samples, which is the main critique of VAE. The visual quality of the generated samples of HBAE is slightly worse than the GAN counterpart in the control experiment setting, but GANs do not directly offer mechanisms to produce reconstructions. There are also known ways of improving the absolute quality of HBAE’s samples, including training bigger models, using larger batch sizes etc, which we leave as future work. In addition, we have tried to train an additional encoder with the trained GAN generator as decoder by minimizing l_1 loss, and showed that this gives much worse reconstruction results than both VAE and HBAE. We have updated the draft to include an Appendix to reflect the aforementioned comparisons (Figure 8&9 in the updated draft; compare to HBAE results in Figure 4 showing much crisper reconstructions and samples).
>
> 3. Results on the face datasets: identity preserving and controlled variation of generation
> HBAE by nature is an unsupervised learning method; when applied to datasets consisting of human faces, it happens to be able to learn reconstructions that vary the input face in certain interesting ways. However, there is very little reason to believe that the reconstructions learned by an HBAE should preserve the identity of a human face, as the notion of identity is not fed into the model in any way, and there are numerous other sources of variation in the datasets. Similarly, we agree that it’d be interesting if the HABE automatically learns a disentangled representation that provides meaningful attributes of human faces, but this is difficult to achieve without explicit supervision. We thus leave both directions as future work, where one can extend HBAE to a semi-supervised setting, achieving desired control over certain factors. These directions will be emphasized in the Discussion section of the updated manuscript.

---

> > ### Author Response · Authors · 2019-11-12
> > **Author's response to Reviewer #1 part 2/2**
> >
> > 4. Quality of the Set HBAE
> > As the reviewer noted, the image set reconstruction and generation task is a novel problem with no prior reference. It is also challenging from the methodology perspective as most of the reconstruction based models would fail, because there is no one-to-one mapping between the set representation and elements within the set. We thus believe that the results we achieve with Set HBAE are significant and non-trivial, as the model clearly learns meaningful information on both the two datasets evaluated in the paper. Evaluation is challenging w.r.t. generative models in general, and this is especially true for our set reconstruction/generation case. However, we have shown quantitatively that the learned representations from Set HBAE shows promising results in the semi-supervised learning case, which serves as additional evidence for the effectiveness of the model.
> >
> > Minors:
> > Figure 1: We are adopting the plate notation [3] to explain the difference between the three models from a graphical model perspective. Here we want to highlight that a VAE assumes that all dimensions x_k of x are conditionally independent given the latent z, while an HBAE models the full interaction between all dimensions of x, with the EBM. The Set-HBAE assumes that the elements (there are M of them) within a set are conditionally independent given z, but the interaction between feature dimensions are still fully modeled. We will make this more explicit in the updated manuscript.
> >
> > Face alignment: Celeba comes with aligned faces, while VGGFace2 does not align them. We use the native forms of the respective datasets.
> >
> >
> > Refs:
> > [1] Stochastic Variational Video Prediction, Babaeizadeh et al
> > [2] World Models, Ha and Schmidhuber
> > [3] https://en.wikipedia.org/wiki/Plate_notation
> > [4] Deep Sets, Zaheer et al
> > [5] Exploiting generative models In discriminative classifiers, Jaakkola and Haussler
> > [6] Improved Techniques for Training GANs, Salimans et al

---

### Official Review · AnonReviewer4 · 2019-12-08
**Official Blind Review #4**

**Rating:** 1

**Review:**

I don't think this paper should be accepted.  In my opinion, the mix of EBM and VAE is not really compelling;  and it is not clear at all to me that one gets much from the "V" in this setting.  Furthermore,  the experimental results are not great either qualitatively (by the standards of generative-models-of-images in 2019) or quantitatively (even by the standards of GAN papers).  Finally, the author's claims about sets seems tacked on, and unrelated to the rest of the paper.  Modern neural networks (attention/transformers/graph-nn, etc...) handle sets naturally, and could be used with any other conditional generative model.  To my eye, the results in e.g. figure 6 do not really seem like the model is matching the set, and the authors make no attempt to formalize or quantify how well their model generates things "in the set".


**Experience Assessment:**

I have published in this field for several years.

**Review Assessment: Checking Correctness Of Derivations And Theory:**

I assessed the sensibility of the derivations and theory.

**Review Assessment: Checking Correctness Of Experiments:**

I assessed the sensibility of the experiments.

**Review Assessment: Thoroughness In Paper Reading:**

I read the paper at least twice and used my best judgement in assessing the paper.

---

### Author Response · Authors · 2019-11-12
**Draft updated**

The draft has been updated with Figure 8 & 9 added to the Appendix. Other parts remain unchanged.

---

### Decision · Program_Chairs · 2019-12-19

**Decision:**

Reject

**Comment:**

This paper introduces a probabilistic generative model which mixes a variational autoencoder (VAE) with an energy based model (EBM). As mentioned by all reviewers (i) the motivation of the model is not well justified (ii) experimental results are not convincing enough. In addition (iii) handling sets is not specific to the proposed approach, and thus claims regarding sets should be revised.